# WHEN FORCES DISAGREE: A DATA-FREE FAST DIAGNOSTIC FROM INTERNAL CONSISTENCY OF DIRECT-FORCE NEURAL NETWORK POTENTIALS

## ABSTRACT

Direct-force neural network potentials (NNIPs) offer superior speed for atomistic simulations, but their reliability is limited by the lack of a fast and data-free uncertainty estimate to monitor the impact of non-conservativity and prediction errors. While ensembles are data-free but slow, and other single-model methods often require training data, we introduce an approach that combines the advantages of both. Our metric is derived from the internal disagreement between a model's directly predicted force and its energy-gradient-derived force, motivated by our finding that a model's internal self-consistency is more critical for algorithmic stability than its external accuracy. We then identify an asymmetric failure mode inherent to the direct-force architecture that this metric can diagnose, and also show a strong monotonic correlation between the disagreement and the true force error across diverse materials and out-of-distribution structures. We propose the link between internal disagreement and practical reliability is a consequence of inter-head influence via the shared graph neural network embedding. We provide direct evidence for this mechanism by showing that fine-tuning the conservative force pathway on adversarial data that maximizes this internal disagreement measurably improves the stability of simulations driven only by the direct force. The metric serves as a versatile and out-of-the-box tool that is competitive with expensive ensembles, offering both an on-the-fly assessment of model reliability and a principled method for generating targeted data to improve the stability of direct-force models.

## 1 INTRODUCTION

Direct-force neural network interatomic potentials (NNIPs) are increasingly favored for their computational efficiency in large-scale atomistic simulations (Gasteiger et al., 2021; Liao et al., 2024; Neumann et al., 2024; Rhodes et al., 2025). This speed, however, comes at the cost of reliability. By decoupling the force prediction from a scalar potential, direct-force models are not guaranteed to be energy-conserving, leading to known algorithmic instabilities in molecular dynamics (MD) (Bigi et al., 2025) and poor performance in property prediction tasks that depend on the potential energy surface (PES) curvature (Póta et al., 2024; Loew et al., 2025). Hybrid integration schemes like Multiple-Time-Stepping (MTS), where conservative forces are used to correct direct forces at a certain frequency during simulations, have been shown to successfully stabilize the simulations while mostly recover the speed of direct forces (Bigi et al., 2025). However, even with such schemes, a lack of a fast and effective metric to monitor a model's reliability (e.g., the impact of non-conservativity and prediction errors) in real-time still remains.

Current uncertainty quantification (UQ) methods present a difficult trade-off for developing such a metric: model ensembles are data-free but computationally prohibitive, while faster single-model methods often require access to the original training data. This work aims develop a universal monitoring metric for direct-force models that combines the advantages of both paradigms. To do so, we first investigate the fundamental principles that govern simulation stability since direct-force models are known for its unstability (Fu et al., 2024; Bigi et al., 2025). While the work of Fu et al. (2023; 2024) established that static force accuracy is an insufficient metric for dynamics and proposed that conservativity are one of the key requirements for reliable NNIPs, the relative importance of

conservativity compared to accuracy has not been directly demonstrated. Our investigation leads to a series of discoveries that provide this missing evidence.

We first provide empirical proof that a model's internal self-consistency is more critical for algorithmic stability than its external accuracy, confirming and building upon the principles laid out by Fu et al. (2023; 2024). This finding motivates our use of an internal disagreement metric, the Force Delta ($U_\Delta$), which is the difference between a model's direct force prediction, $\hat{\mathbf{F}}_{nc}$, and its internally self-consistent, energy-derived force, $\hat{\mathbf{F}}_c$. Using this metric, we then identify an Asymmetric Failure Mechanism inherent to the pre-trained dual-output direct-force architecture. We also find that $U_\Delta$ is a more consistent predictor of instability than the direct force error magnitude against references alone. We propose the link between this internal metric and practical reliability is a consequence of inter-head influence via the shared GNN embedding, and we provide direct evidence for this mechanism by demonstrating that fine-tuning the conservative force pathway measurably improves the stability of simulations driven only by the direct force.

Our contributions are as follows:

- We provide the first direct, experimental proof that for simulation stability, a model's internal self-consistency is more critical than its external accuracy.
- We introduce the Force Delta ($U_\Delta$) as a versatile, data-free UQ metric that identifies an Asymmetric Failure Mechanism inherent to direct-force architectures with competitive performances with expensive ensembles.
- We provide three-layered evidence for inter-head influence in direct-force models: (1) correlational evidence where the magnitude of the error in the two forces (one from the energy and the other from the force head) is correlated and captured by $U_\Delta$, (2) predictive evidence where the magnitude of $\hat{\mathbf{F}}_c$'s error predicts the pathological character of the $\hat{\mathbf{F}}_{nc}$'s error that causes energy drift, and (3) causal evidence where finetuning $\hat{\mathbf{F}}_c$ improves $\hat{\mathbf{F}}_{nc}$'s stability.
- We present a complete workflow, using $U_\Delta$ to generate targeted data to iteratively improve the stability of both pre-trained and already fine-tuned direct-force models.

## 2 BACKGROUND AND RELATED WORK

**Machine Learning Interatomic Potential**    Machine Learning Interatomic Potentials (MLIPs) aim to approximate the quantum mechanical potential energy surface (PES) with the efficiency of classical force fields. Early influential models were descriptor-based, first mapping local atomic environments to a set of fixed, hand-crafted feature vectors (descriptors) which were then fed into a simple machine learning model. Seminal examples in this class include Behler-Parrinello Neural Networks (Behler & Parrinello, 2007), Gaussian Approximation Potentials (GAP) (Bartók et al., 2010), Spectral Neighbor Analysis Potentials (SNAP) (Thompson et al., 2015), and Moment Tensor Potentials (MTP) (Shapeev, 2016).

A subsequent generation of models moved towards end-to-end deep learning, using neural networks to learn the feature representation directly from atomic coordinates. Architectures like ANI (Smith et al., 2017) and SchNet (Schütt, 2017) were foundational in this area, often building upon the message-passing framework of Graph Neural Networks (GNNs) (Gilmer, 2017; Battaglia, 2018). The current state-of-the-art is dominated by E(3)-equivariant GNNs, which build in physical symmetries (roto-translational equivariance) directly into the network architecture. This inductive bias significantly improves data efficiency and generalization (Musil, 2021). Foundational equivariant architectures include Tensor Field Networks (Thomas, 2018), NequIP (Batzner et al., 2022), MACE (Batatia, 2022), and Allegro (Musaelian, 2023). The success of these models has spurred the development of large-scale, pre-trained "foundation models" for atomistic simulation, such as CHGNet (Deng, 2023) and the direct-force models used in this work.

**Direct-Force Architectures.**    The drive for computational efficiency has popularized direct-force architectures in many state-of-the-art models (Gasteiger et al., 2021; Batatia, 2022; Liao et al., 2024; Neumann et al., 2024; Rhodes et al., 2025). In contrast to conservative models, these architectures predict atomic forces as a direct, equivariant vector output of the GNN, rather than computing

them as the gradient of a predicted scalar energy. This approach can yield significant performance benefits, including faster training and inference and lower memory usage, as it often avoids the computational cost of a backward pass (i.e., backpropagation) through the network (Gasteiger et al., 2021). Architecturally, these models typically use a shared GNN encoder to generate atomic representations, which are then fed to separate output heads. The first head predicts the direct, non-conservative force, $\hat{\mathbf{F}}_{\text{nc}}$, as a direct equivariant vector output. The second head predicts a scalar energy, $\hat{E}$. The conservative force, $\hat{\mathbf{F}}_{\text{c}}$, is the gradient of this energy, $\hat{\mathbf{F}}_{\text{c}} = -\nabla_{\mathbf{R}}\hat{E}$.

A key assumption of the direct-force paradigm is that the faster $\hat{\mathbf{F}}_{\text{nc}}$ can serve as a sufficient substitute for the computationally more expensive $\hat{\mathbf{F}}_{\text{c}}$, which requires a backward pass through the network. Consequently, the conservative force pathway is typically ignored during training. The model is instead trained by minimizing a joint loss function on the energy and the direct forces, which encourages both accuracy on Density Functional Theory (DFT) targets and, implicitly, consistency between the two pathways:

$$\mathcal{L} = \lambda_E \mathcal{L}_E(\hat{E}, E_{\text{DFT}}) + \lambda_F \mathcal{L}_F(\hat{\mathbf{F}}_{\text{nc}}, \mathbf{F}_{\text{DFT}}) \tag{1}$$

**The Consequences of Non-Conservativity.** The efficiency gain of using $\hat{\mathbf{F}}_{\text{nc}}$ comes at the expense of guaranteed energy conservation (Chmiela, 2017). This lack of an underlying potential violates the assumptions of algorithms that navigate the Potential Energy Surface (PES). Symplectic integrators used in MD assume forces are the exact gradient of a potential to conserve the Hamiltonian (Hairer, 2006; Leimkuhler & Reich, 2004; Tuckerman, 2023). Non-conservative forces lead to unphysical energy drift and instabilities in NVE simulations due to its nature of not being an exact spatial gradient of any potential (Bigi et al., 2025; Fu et al., 2024). This non-conservativity also creates artifacts in NVT simulations that are difficult or impossible to correct with thermostats without disrupting dynamical or structural properties Bigi et al. (2025). Similarly, gradient-based optimizers require a consistent PES for stable convergence (Nocedal & Wright, 2006), leading to more fragile geometry optimization using non-conservative forces compared to conservative forces Bigi et al. (2025).

**Requirements for Stable and Accurate MLIPs** A growing body of work has established that the requirements for a reliable MLIP go far beyond simple accuracy on a static test set. The seminal work of Fu et al. (2023) provided the first large-scale benchmark demonstrating that static force error is often an insufficient metric for predicting the dynamic stability of a simulation. This exposed a critical gap between how models are benchmarked and how they are used in practice. However, a clear and direct experimental validation of the relative importance of conservativity and accuracy against a DFT reference is still lacking. By designing experiments that isolate the effects of accuracy from self-consistency, we provide the first direct, quantitative evidence for the relative importance of conservativity compared to accuracy.

**The Limitations of Existing UQ Methods.** Quantifying uncertainty is crucial for monitoring reliability (Abdar et al., 2021; Musil et al., 2023). Deep ensembles remain the standard for epistemic uncertainty (Lakshminarayanan et al., 2017), but their high computational cost (typically 5-10x) is prohibitive for large-scale simulations (Wen & Tadmor, 2020). Single-model Bayesian approaches (Gal & Ghahramani, 2016; Vandermause et al., 2020) often require modified training. Data-dependent methods (e.g., distance in latent space) (Hirschfeld et al., 2020; Podryabinkin & Shapeev, 2017) are unsuitable for foundation models as they require access to massive datasets and can perform poorly on heterogeneous data (Tan et al., 2023; Jablonka et al., 2021; Wang et al., 2023). A fast, data-free metric derived from the model's internal physics is needed.

## 3 METHODS

**Force Definitions and Metrics.** A direct-force NNIP provides two distinct force predictions. The first is the direct, non-conservative force, $\hat{\mathbf{F}}_{\text{nc}}$, which is the direct equivariant vector output of the model's force head. The second is the conservative force, $\hat{\mathbf{F}}_{\text{c}}$, which is derived from the model's own learned potential energy surface, $\hat{E}$, via the chain rule: $\hat{\mathbf{F}}_{\text{c}} = -\nabla_{\mathbf{R}}\hat{E}$. We define our primary

diagnostic, the **Force Delta** ($U_\Delta$), as the root-mean-square difference between these two predictions:

$$U_\Delta(\mathbf{R}) = \sqrt{\frac{1}{3N} \sum_{i=1}^{N} \|\hat{\mathbf{F}}_{\text{nc},i}(\mathbf{R}) - \hat{\mathbf{F}}_{\text{c},i}(\mathbf{R})\|^2} \tag{2}$$

To validate this metric, we compare it against two true error metrics calculated with respect to a ground-truth DFT force, $\mathbf{F}_{\text{DFT}}$. The non-conservative error is $\varepsilon_{\text{nc}} = \sqrt{\frac{1}{3N} \sum_{i=1}^{N} \|\hat{\mathbf{F}}_{\text{nc},i}(\mathbf{R}) - \mathbf{F}_{\text{DFT},i}(\mathbf{R})\|^2}$, and the conservative error is $\varepsilon_{\text{c}} = \sqrt{\frac{1}{3N} \sum_{i=1}^{N} \|\hat{\mathbf{F}}_{\text{c},i}(\mathbf{R}) - \mathbf{F}_{\text{DFT},i}(\mathbf{R})\|^2}$.

**Models and Systems.** We use a suite of publicly available, pre-trainead direct-force models, primarily from the Orb (Neumann et al., 2024; Rhodes et al., 2025) and EquiformerV2 (Liao et al., 2024) families. Our test set includes a diverse range of systems, including crystalline solids (e.g., ice, LGPS, $Mg_{17}Al_{12}$), a liquid water box, surface, and molecules, designed to probe model performance on both in- and out-of-distribution structures. DFT calculations for ground-truth forces were performed with VASP using the PBE functional. Further details on all models, systems, and DFT parameters are in the Appendix.

**Simulation Protocols.** Molecular dynamics simulations were performed in both the microcanonical (NVE) and canonical (NVT) ensembles using the Velocity Verlet integrator. NVE simulations were used to assess energy conservation and drift, while NVT simulations were used to test for other artifacts, such as temperature fluctuations. Further details are in the Appendix

**Adversarial Generation of OOD Structures.** To efficiently generate challenging OOD configurations, we employ a differentiable adversarial attack (Schwalbe-Koda et al., 2021). Starting from equilibrium structures, we iteratively perturb the atomic positions $\mathbf{r}$ to find configurations that are both physically plausible (low energy) and maximally inconsistent. This is achieved by updating positions along a composite gradient that simultaneously maximizes our diagnostic, $U_\Delta$, while minimizing the predicted energy, $\hat{E}$:

$$\mathbf{r}_{\text{new}} = \mathbf{r}_{\text{old}} + \alpha \nabla_{\mathbf{r}} U_\Delta - \beta \nabla_{\mathbf{r}} \hat{E} \tag{3}$$

where $\alpha$ and $\beta$ are the respective learning rates. This process efficiently drives the system towards high-uncertainty, low-energy regions where the model's internal physics is most stressed.

## 4 RESULTS

Our results are presented in four parts. We first experimentally establish that for stable simulations, a model's self-consistency is more critical than its external accuracy. We then introduce the Force Delta, $U_\Delta$, use it to identify the asymmetric failure mode, and validate it as a robust diagnostic for both conservative and direct force errors. Building on this, we show that $U_\Delta$ is a more consistent indicator of algorithmic instability than standard error metrics. Finally, we provide final direct evidence for the underlying inter-head mechanism by using $U_\Delta$-maximized data in fine-tuning experiments to demonstrably improve model stability.

### 4.1 EXPERIMENTAL INVESTIGATION OF SIMULATION STABILITY REQUIREMENTS

We employ DFT calculations to quantitatively investigate the relative importance of accuracy against $\mathbf{F}_{\text{DFT}}$ compared to conservativity during the energy drift in NVE simulations. To isolate the effects of accuracy from self-consistency, we perform a series of NVE simulations on a liquid water box, with consistent findings for other systems presented in the Appendix. The results, shown in Figure 1, reveal a clear hierarchy. First, we compare a simulation driven by the accurate but non-conservative force ($\hat{\mathbf{F}}_{\text{nc}}$) of the `orb-v3-direct-inf-mpa` model to one driven by its less accurate but internally self-consistent conservative force ($\hat{\mathbf{F}}_{\text{c}}$) obtained by backpropagating predicted energy to obtain its negative spatial derivative of the same model. The $\hat{\mathbf{F}}_{\text{nc}}$-driven simulation is unstable, while the $\hat{\mathbf{F}}_{\text{c}}$-driven run is perfectly stable, providing direct proof that self-consistency (i.e., forces being an

exact gradient of model's predicted energy) is more critical than accuracy. This confirms that the small error of $\hat{\mathbf{F}}_{\mathrm{nc}}$ against $\mathbf{F}_{\mathrm{DFT}}$ during NVE accumulates and causes the drift. No matter how close $\hat{\mathbf{F}}_{\mathrm{nc}}$ is to an exact gradient of DFT energy ($\mathbf{F}_{\mathrm{DFT}}$), it will never be an exact gradient of any potential and therefore produces artifact. On the other hand, $\hat{\mathbf{F}}_{\mathrm{c}}$ of `orb-v3-direct-inf-mpa`, while being inaccurate compared to $\mathbf{F}_{\mathrm{DFT}}$, produces a stable simulations since it satisfies the symplectic requirements of the integrator by being an exact gradient of its own predicted energy. In other words, $\hat{\mathbf{F}}_{\mathrm{c}}$ is self-consistent with its own potential energy landscape.

The effects of energy drift also leads to larger temperature fluctuations compared to simulations with inaccurate $\hat{\mathbf{F}}_{\mathrm{c}}$ in NVT as shown in the Appendix. Bigi et al. (2025) have demonstrated that this artifact in NVT is difficult or impossible to contain using a thermostat without disrupting dynamical properties. This finding provides direct evidence for the necessity of conservativity and the fundamental justification for hybrid integration schemes, such as the Multiple-Time-Stepping (MTS) method proposed by Bigi et al. (2025), which leverage the stability of the conservative force to correct the trajectory of the direct force. Furthermore, these experiments establish the scientific motivation for a diagnostic that can probe these internal model properties.

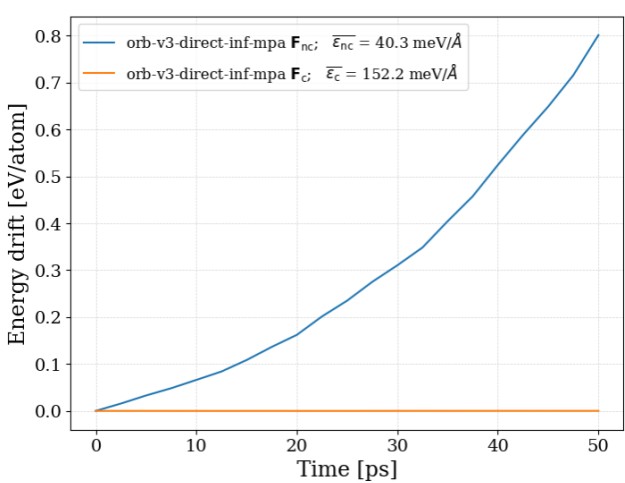

Figure 1: Energy evolution during NVE simulations of a liquid water box (15 Å side length). The run driven by the self-consistent and smooth but less accurate $\hat{\mathbf{F}}_{\mathrm{c}}$ (green) is stable. In contrast, the run driven by the more accurate but non-conservative $\hat{\mathbf{F}}_{\mathrm{nc}}$ (orange) is unstable.

### 4.2 THE FORCE DELTA: A DIAGNOSTIC FOR FORCE ERRORS

Having established the requirements for simulation stability, we now validate the Force Delta, $U_\Delta$, as a diagnostic for the model properties that govern these principles. A key to understanding $U_\Delta$'s utility is the inherent asymmetry in how the two force predictions are generated and supervised.

#### 4.2.1 THE ASYMMETRIC FAILURE MECHANISM

The dual-output architecture of direct-force models leads to a predictable, asymmetric failure mode when the model is pushed out-of-distribution (OOD). This arises from two factors: asymmetric supervision and the mathematical properties of differentiation. The direct force, $\hat{\mathbf{F}}_{\mathrm{nc}}$, is strongly regularized by direct supervision on vector force DFT data. In contrast, the model's energy, $\hat{E}$, is only weakly supervised by scalar values, which is insufficient to regularize the curvature of the potential energy surface. Moreover, mathematically, differentiation acts as a high-pass filter, meaning that any small and high-frequency "ripples" in the under-regularized $\hat{E}$ (i.e., non-smooth curvature) are amplified into large-magnitude errors in its gradient, $\hat{\mathbf{F}}_{\mathrm{c}}$.

This mechanism predicts that as a model goes OOD, the error in the conservative force, $\varepsilon_{\mathrm{c}}$, should grow much more rapidly than the error in the non-conservative force, $\varepsilon_{\mathrm{nc}}$. This large discrepancy in

error magnitudes is the key to understanding the utility of the Force Delta. Since $U_\Delta = \|\hat{\mathbf{F}}_{\mathrm{nc}} - \hat{\mathbf{F}}_{\mathrm{c}}\|$, it can be rewritten in terms of the respective error vectors as $U_\Delta = \|\vec{\varepsilon}_{\mathrm{nc}} - \vec{\varepsilon}_{\mathrm{c}}\|$. When the conservative error dominates such that $\|\vec{\varepsilon}_{\mathrm{c}}\| \gg \|\vec{\varepsilon}_{\mathrm{nc}}\|$, the smaller term becomes negligible, and the expression simplifies to $U_\Delta \approx \| - \vec{\varepsilon}_{\mathrm{c}}\| = \varepsilon_{\mathrm{c}}$. The Force Delta thus becomes a direct and precise mathematical proxy for the error in the conservative force. We test this by using adversarial attacks to efficiently generate OOD structures. As shown in Figure 2a, we observe a strong correlation between $U_\Delta$ and $\varepsilon_{\mathrm{c}}$ for the `orb-v3-direct-20-mpa` model on several crystalline systems. This result, which is consistent across tested models in the Orb and EqV2 families and most systems for OOD structures from both adversarial attack and high runaway temperatures during NVE (see Appendix), provides empirical evidence for the asymmetric failure mechanism and establishes $U_\Delta$ as a reliable probe of the model's internal physical breakdown.

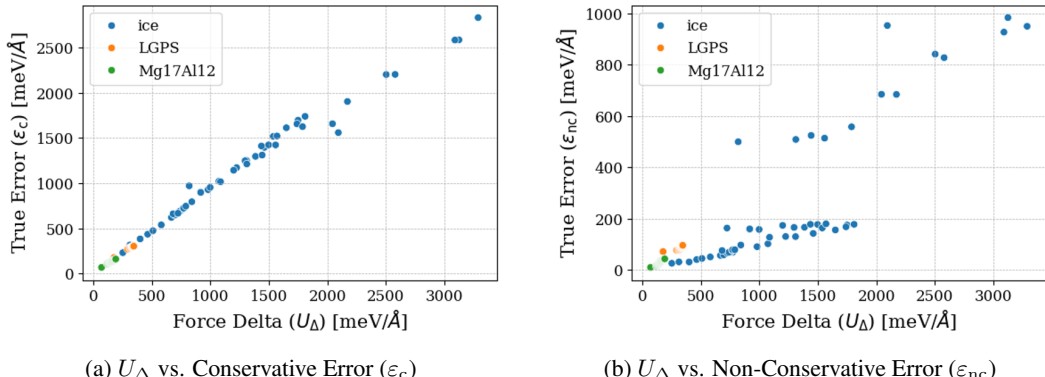

(a) $U_\Delta$ vs. Conservative Error ($\varepsilon_{\mathrm{c}}$)      (b) $U_\Delta$ vs. Non-Conservative Error ($\varepsilon_{\mathrm{nc}}$)

Figure 2: The Force Delta ($U_\Delta$) as a robust indicator of force errors for out-of-distribution structures generated via an adversarial attack on the `orb-v3-direct-20-mpa` model. Initial configurations were obtained from Materials Project (Jain et al., 2013) and geometrically-optimized using the model (a) $U_\Delta$ shows a near-perfect correlation with the conservative force error, $\varepsilon_{\mathrm{c}}$. The Spearman's rank correlation for ice ($r_s = 0.99$), LGPS ($r_s = 0.88$), and $\mathrm{Mg}_{17}\mathrm{Al}_{12}$ ($r_s = 1.00$) demonstrates the asymmetric failure mechanism. (b) $U_\Delta$ also maintains a strong positive correlation (ice ($r_s = 0.91$), LGPS ($r_s = 1.00$), and $\mathrm{Mg}_{17}\mathrm{Al}_{12}$ ($r_s = 1.00$)) with the direct, non-conservative force error, $\varepsilon_{\mathrm{nc}}$, establishing its utility as a general UQ metric.

### 4.2.2 FORCE DELTA AS A GENERAL UQ METRIC FOR FORCE ERROR

After demonstrating $U_\Delta$ is a strong indicator of the model's internal conservative force error, we now investigate its utility as a practical UQ metric for the non-conservative force error that is used due to its superior inference speed, $\varepsilon_{\mathrm{nc}}$. A strong correlation between these two errors would suggest a deeper connection between the model's two prediction heads. We first test this relationship on a diverse benchmark set of ten systems, including crystalline solids, surface, and molecules to represent both in- and out-of-distribution data for the pre-trained models (see Appendix for dataset and model details). As shown in Table 1, the single-model $U_\Delta$ exhibits a strong correlation with $\varepsilon_{\mathrm{nc}}$, and its performance is competitive with, or superior to, the expensive multi-model ensemble variance of force predictions, $U_{\mathrm{var}}$. It is crucial to note that these ad-hoc collection of models are not "deep ensembles" in the strictest sense, as they were not co-trained with varied initializations on an identical dataset. However, they represent the most direct ensemble-based UQ approach available to a user working with publicly available, pre-trained models. Furthermore, combining all 12 models into a single ensemble would be physically and statistically invalid. The variance would be dominated by the systematic bias between the different ground-truth DFT methods used for training (DFT vs. DFT+D3) rather than true epistemic uncertainty.

To further test the robustness of this relationship in high-uncertainty regimes, we analyzed the correlation on OOD structures generated via our adversarial attack. As shown for a representative model in Figure 2b, $U_\Delta$ maintains a strong positive correlation with $\varepsilon_{\mathrm{nc}}$, even as the model is pushed far from its training distribution. This result is consistent across most models and systems we tested. For the `eqV2-dens-31M-mp` model, the correlations between $U_\Delta$ and $\varepsilon_{\mathrm{nc}}$ appear to be weaker than the orb models. This could be attributed to the different relative weights between each head

($\lambda_F : \lambda_E$ ratio) between each model family. The ratio is 1 for orb and 25 for eqV2, hence the $\hat{\mathbf{F}}_{\mathrm{nc}}$ of the eqV2 model being more robust and the resulting weaker correlations (see Appendix for full correlation tables).

For a few specific systems (e.g., MoF5, aspirin), the correlation is weak or even negative. This is because the true error of the initial equilibrium structure was already substantial (see Appendix). Consequently, the adversarial attack, while still finding high-uncertainty configurations, did not produce as dramatic an increase in error, which can weaken the calculated correlation coefficient. Crucially, the Force Delta for these points is consistently high, correctly flagging them as unreliable. This shows the metric functions as an effective "failure detector" for applications like active learning or molecular dynamics (MD) monitoring, where identifying failure is often more important than perfect error prediction.

Since $U_\Delta$ has a near-perfect correlation with the conservative force error ($\varepsilon_{\mathrm{c}}$), this means that $\varepsilon_{\mathrm{c}}$ is a reliable indicator of the direct force head's prediction error ($\varepsilon_{\mathrm{nc}}$). This finding provides the first, correlational evidence for inter-head influence, where the state of one prediction pathway ($\hat{E}$ which gives $\hat{\mathbf{F}}_{\mathrm{c}}$) informs on the other ($\hat{\mathbf{F}}_{\mathrm{nc}}$), all captured by their disagreement $U_\Delta$. This establishes $U_\Delta$ as a reliable and robust UQ metric for the direct force predictions used in a wide range of applications such as geometry optimization and property prediction.

Table 1: Spearman's rank correlation ($r_s$) comparing the single-model $U_\Delta$ against the ad-hoc ensemble variance $U_{\mathrm{var}}$ as predictors of the non-conservative force error, $\varepsilon_{\mathrm{nc}}$, on a diverse benchmark set. The reported single-model's $r_s$ value is the average of $r_s$ between $U_\Delta$ and $\varepsilon_{\mathrm{nc}}$ on the ten systems over models. Full details are in the Appendix.

| Model Family | Avg. $r_s$ (Single-Model $U_\Delta$) | $r_s$ (Ensemble $U_{\mathrm{var}}$) | Relative Cost |
|---|---|---|---|
| Orb (5 models) | $0.70 \pm 0.04$ | **0.73** | $\approx 5\times$ |
| EquiformerV2 (7 models) | **$0.91 \pm 0.02$** | 0.79 | $\approx 7\times$ |

### 4.3 $U_\Delta$ AS A CONSISTENT INDICATOR OF ALGORITHMIC INSTABILITY

We now test the ability of $U_\Delta$ to diagnose the practical consequence of these errors: algorithmic instability on four systems (ice, LGPS, $Mg_{17}Al_{12}$, and water box). While the accumulated error, $\varepsilon_{\mathrm{nc}}$, is the direct physical cause of energy drift in NVE simulations, we find its predictive power is complex and model-dependent. For models in a fragile state where $\varepsilon_{\mathrm{nc}}$ becomes large (e.g., orb-d3-xs-v2), its magnitude correlates well with drift and all metrics ($\varepsilon_{\mathrm{nc}}, \varepsilon_{\mathrm{c}}, U_\Delta$, and energy drift) are well-correlated (see Appendix). However, for robust, pre-trained models, $\varepsilon_{\mathrm{nc}}$ often operates in a low-error regime where its correlation with instability is not guaranteed.

Our results reveal a clear distinction between the predictive power of the external error magnitude and the internal inconsistency in this critical low-error regime. For strongly-regularized models like the OrbV3 family, the correlation between the average $\varepsilon_{\mathrm{nc}}$ during a simulation and the resulting energy drift is weak and noisy, as shown in Figure 3a. In contrast, Figure 3b shows that the average Force Delta, $U_\Delta$, exhibits a more consistent positive correlation with the energy drift. For other models like eqV2-dens-31M-mp, all metrics happen to be well-correlated (see Appendix for a full analysis). This demonstrates that while the predictive power of $\varepsilon_{\mathrm{nc}}$'s magnitude is inconsistent across different model architectures and training regimes, the internal inconsistency, $U_\Delta$, is a more reliable indicator of the pathological character of the error that governs the severity of the instability, an artifact difficult to monitor and suppress without disrupting dynamical and structural properties in NVT simulations (Bigi et al., 2025). In fact, the mean error of the conservative forces ($\varepsilon_{\mathrm{c}}$) has the strongest correlation ($r_s = 0.97$) with the energy drift in simulations performed using $\hat{\mathbf{F}}_{\mathrm{nc}}$. Since $U_\Delta$ has a strong correlation with $\varepsilon_{\mathrm{c}}$ (also with $r_s = 0.97$), it also predicts energy drift as well as $\varepsilon_{\mathrm{c}}$. This provides predictive evidence for inter-head influence, where the state of the model's internal physics and the error of the other (energy) head is a more consistent probe of its reliability than the accuracy of the head used in the simulations ($\hat{\mathbf{F}}_{\mathrm{nc}}$) alone.

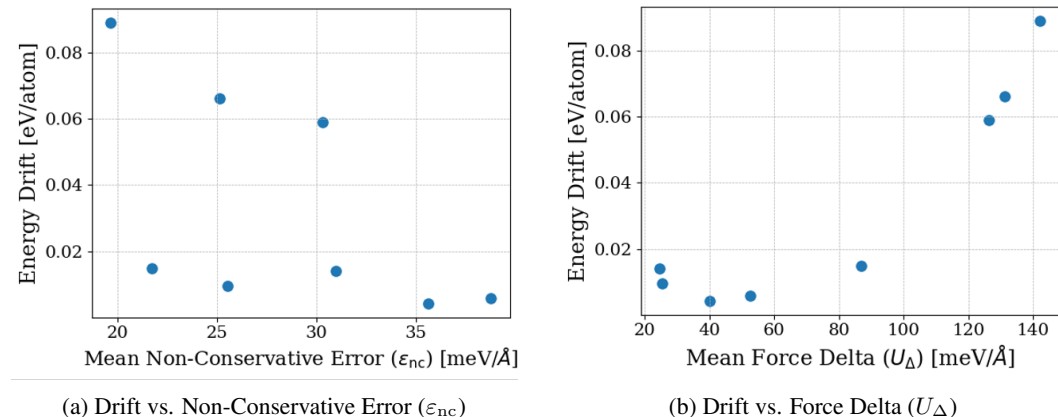

(a) Drift vs. Non-Conservative Error ($\varepsilon_{\mathrm{nc}}$)  (b) Drift vs. Force Delta ($U_\Delta$)

Figure 3: The Force Delta ($U_\Delta$) as a consistent indicator of algorithmic instability for robust direct OrbV3 (`orb-v3-direct-inf-mpa` and `orb-v3-direct-20-mpa`) models. Each point represents a 10-ps NVE simulation. (a) In the low-error regime, the magnitude of the non-conservative force error, $\varepsilon_{\mathrm{nc}}$, shows a weak and noisy correlation with the total energy drift ($r_s = 0.24$). (b) In the same set of simulations, the internal inconsistency, $U_\Delta$, shows a more consistent positive correlation with the energy drift ($r_s = 0.91$). A full analysis of all models is in the Appendix.

## 4.4 STABILITY-IMPROVING FINETUNING EXPERIMENTS

The strong correlations observed between the model's internal inconsistency and its practical reliability suggest a deep connection between the energy and force prediction heads. We propose that this inter-head influence is mediated by the shared GNN embedding. To provide direct, causal evidence for this mechanism, we perform a series of fine-tuning experiments using data generated from our adversarial attack on $U_\Delta$. If our hypothesis is correct, then improving the internal physics of the model by training its conservative pathway should have a direct, measurable effect on the stability of simulations driven only by the direct $\hat{\mathbf{F}}_{\mathrm{nc}}$ force. Note that we have to always finetune the direct force head $\hat{\mathbf{F}}_{\mathrm{nc}}$ here to avoid degradation since the pre-trained models were trained on $\hat{\mathbf{F}}_{\mathrm{nc}}$ and the simulations are typically performed using $\hat{\mathbf{F}}_{\mathrm{nc}}$.

We test this on the `orb-v3-direct-inf-mpa` model with the ice system. As shown in Figure 4, finetuning the model on just 100 adversarial structures leads to a clear and stepwise improvement in stability. Fine-tuning only the $\hat{\mathbf{F}}_{\mathrm{nc}}$ head reduces the energy drift compared to the pre-trained baseline. Critically, fine-tuning both the $\hat{\mathbf{F}}_{\mathrm{c}}$ and $\hat{\mathbf{F}}_{\mathrm{nc}}$ heads (i.e., conservative fine-tuning) further reduces the drift in the $\hat{\mathbf{F}}_{\mathrm{nc}}$-driven simulation. This provides direct evidence that improving the quality of the model's internal energy landscape improves the non-conservative character of the direct forces. Furthermore, we demonstrate the iterative utility of our method by performing a second adversarial attack on this improved model to generate a new set of 100 structures. Fine-tuning on this "2nd generation" data nearly eliminates the long-term energy drift. These results confirm the inter-head influence mechanism and validate our method as a principled way to generate targeted data for improving model's stability.

## 5 DISCUSSION

Our results, from the error correlations to finetuning experiments, all point to a single unifying mechanism: inter-head influence via the shared GNN embedding. The state of the model's internal physics, represented by $\hat{\mathbf{F}}_{\mathrm{c}}$, is not independent of the direct force prediction, $\hat{\mathbf{F}}_{\mathrm{nc}}$. This finding is consistent with and provides a deeper explanation for previous observations that pre-training a model on its direct-force head provides a more effective starting point for subsequently training the conservative-force pathway (Bigi et al., 2025; Fu et al., 2024). Our work reveals both the diagnostic and improvement sides of this phenomenon: the observable state of one head is a sensitive probe

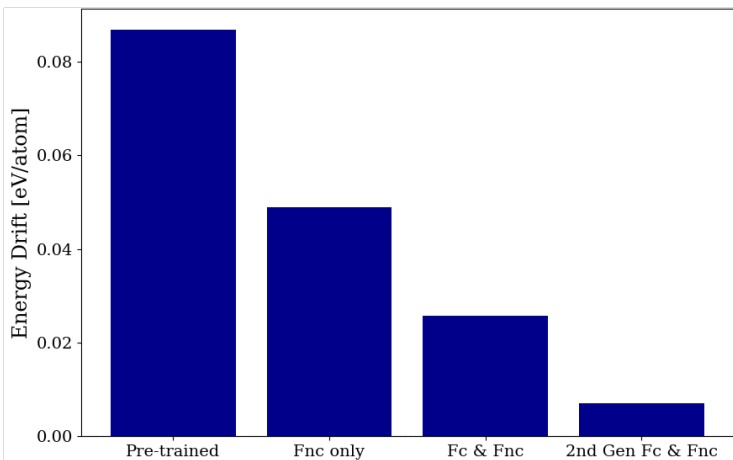

Figure 4: Stepwise reduction in NVE energy drift for the `orb-v3-direct-inf-mpa` model on the ice system after fine-tuning on $U_\Delta$-maximized data. Each bar represents the total drift after 10 ps. Fine-tuning both heads ($\hat{\mathbf{F}}_c$ & $\hat{\mathbf{F}}_{nc}$) is more effective than fine-tuning $\hat{\mathbf{F}}_{nc}$ alone, and a second generation of fine-tuning provides further improvement. Consistent results for another model are in the Appendix.

of the hidden, pathological character of the error in the other, and finetuning one head improves the performance of the other head.

This understanding establishes the Force Delta, $U_\Delta$, which measures the disagreement between two heads as a versatile tool for improving the reliability of the entire workflow of direct-force MLIPs. If the two forces do not agree, at least one of them is wrong against DFT, then the error of the other force could also be large in magnitude as shown by correlational evidence in Section 4.2.2, or could have large non-conservative character that causes artifacts as shown by predictive evidence in Section 4.3 or both. It serves as an on-the-fly monitor for MD simulations (both NVE and NVT) to detect the onset of non-conservative artifacts that can corrupt dynamical properties. For geometry optimizations and property predictions, it acts as a fast, data-free prerequisite check on the trustworthiness of the underlying PES. Furthermore, as our fine-tuning experiments demonstrate, it provides a data-efficient method for generating targeted OOD structures to improve the stability of both pre-trained and already fine-tuned models.

The demonstrated stability improvements from our fine-tuning workflow have direct implications for advanced simulation methods. The Multiple-Time-Stepping (MTS) scheme proposed by Bigi et al. (2025), for instance, relies on the accuracy of both the direct force $\hat{\mathbf{F}}_{nc}$ and the corrective conservative force $\hat{\mathbf{F}}_c$. Our work establishes the Force Delta, $U_\Delta$, as an essential real-time monitor for this scheme, as a large $U_\Delta$ signals that at least one of these forces has become unreliable. Furthermore, by using our method to create a more stable base model, we can logically infer that the MTS algorithm would require less frequent corrective steps. This would lead to a significant increase in the overall simulation speed without sacrificing stability, directly addressing a key challenge in the field.

Finally, it is essential to acknowledge the limitations of this approach, which also point to future directions. Due to its disagreement-based nature, $U_\Delta$ cannot detect "consensus failures" where both force predictions are concurrently wrong, a rare but possible scenario we observed for a specific model-system pair (MoF5) in the Appendix. Furthermore, in the highly consistent, low-error regime, the magnitude of all metrics ($U_\Delta$, $\varepsilon_c$, and $\varepsilon_{nc}$) is small, and correlations with instability may be dominated by numerical noise; differentiating between near-zero uncertainty and noise remains a challenge for any metric. Lastly, $U_\Delta$ is suitable for *ranking* uncertainty, which is the primary requirement for failure detection and active learning. However, it is not *calibrated*; the magnitude of $U_\Delta$ is not a direct predictor of the magnitude of $\varepsilon_{nc}$. Calibrated error estimation would still require system-specific validation (Kuleshov et al., 2018) and represents a key area for future work.

## LLM Usage

For the paper, an LLM was employed solely as a grammar-checking and writing refinement tool. Its use was limited to improving the clarity and coherence of the written language.

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

# A    APPENDIX

## A.1    COMPUTATIONAL DETAILS

### A.1.1    MODEL DETAILS

We utilized 12 pre-trained NNIPs, all of which have the MPTraj data as part of their training dataset. This is to ensure all crystalline solids in the benchmark test set used to compare ad-hoc ensemble's UQ ($U_{\mathrm{var}}$) represent in-distribution of all models. The models spanning two major families of equivariant GNN architectures:

ORB MODELS

The five Orb models (Neumann et al., 2024; Rhodes et al., 2025) used were:

- `orb-d3-xs-v2`
- `orb-d3-v2`
- `orb-d3-sm-v2`
- `orb-v3-direct-inf-mpa`
- `orb-v3-direct-20-mpa`

EQUIFORMERV2 MODELS

The seven EquiformerV2 models (Liao et al., 2024) used were:

- `eqV2_dens_31M_mp`
- `eqV2_dens_153M_mp`
- `eqV2_dens_86M_mp`
- `eqV2_31M_mp`
- `eqV2_31M_omat_mp_salex`
- `eqV2_153M_omat_mp_salex`
- `eqV2_86M_omat_mp_salex`

Detailed studies often used `orb-v3-direct-20-mpa` and `eqV2_dens_31M_mp` as representatives.

### A.1.2 MATERIALS DETAILS

Our test set comprised 10 systems spanning solids ($Mg_{17}Al_{12}$, LGPS, ice, and MoF-5) taken from Materials Project (Jain et al., 2013), surface ($CaPd-NH_2$) taken from OC22 (Tran et al., 2023), and molecules (Ac-Ala3-NHMe, stachyose, aspirin, paracetamol, and DHA taken from the MD22 dataset (Chmiela et al., 2023). The test set for benchmarking $U_\Delta$ against ensemble's $U_{var}$ were taken directly from the mentioned publicly available databases.

### A.1.3 DFT CALCULATION DETAILS

All ground-truth Density Functional Theory (DFT) calculations were performed with the Vienna Ab initio Simulation Package (VASP) (Kresse & Furthmüller, 1996). We used the PBE exchange-correlation functional (Perdew et al., 1996). Calculation parameters were consistent with Materials Project protocols (Jain et al., 2013; Munro et al., 2020).

### A.1.4 MD SIMULATION DETAILS

NVE simulations were performed using the Atomic Simulation Environment (ASE) (Larsen et al., 2017). We used the Velocity Verlet integrator with a timestep of $0.5$ fs for all simulations. For each system-model pair, the initial configuration taken from a corresponding database, relaxed through geometry optimization with force threshold 0.05 eV/Å using the model, and finally equilibrated at NVT 300 K (except for ice which is equlibrated at 200 K). The Nose-Hoover thermostat with `ttime = 10` fs was used to contain temperature fluctuation from non-conservative artifacts Bigi et al. (2025). All NVE simulations start from the last frame of the corresponding NVT-equilibrated frames.

## A.2 ADDITIONAL RESULTS

### A.2.1 ADDITIONAL RESULTS FOR SIMULATION STABILITY REQUIREMENTS

This section provides supplementary results that demonstrate the generality of the findings presented in Section 4.1.

**NVE Simulations on Additional Systems.** The hierarchy of stability requirements observed for the liquid water box holds for other systems. Figure 5 shows the results of equivalent NVE simulations for ice and $Mg_{17}Al_{12}$ crystalline structures using the same set of Orb models. In both cases, the simulation driven by the self-consistent and smooth $\hat{\mathbf{F}}_c$ of the `orb-v3-direct-inf-mpa` model is the most stable. The simulation driven by the more accurate but non-conservative $\hat{\mathbf{F}}_{nc}$ of the same model exhibits significant energy drift.

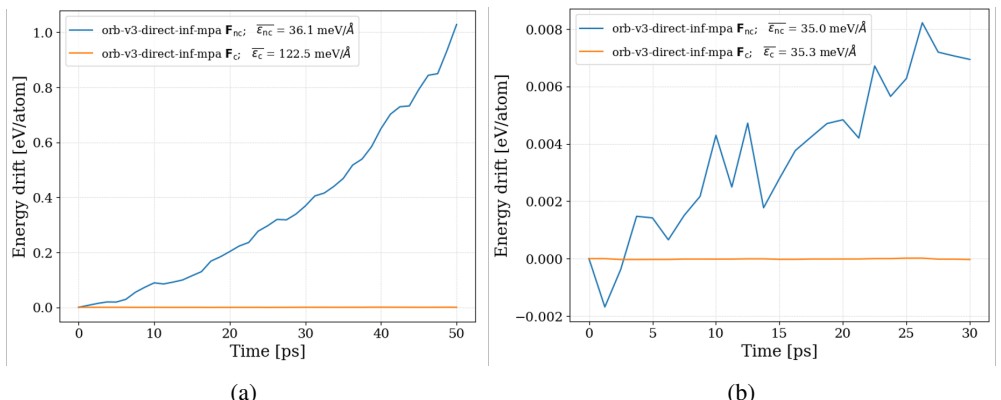

(a)                                                            (b)

Figure 5: Energy evolution during NVE simulations, confirming the stability requirements for (a) an ice crystal and (b) a $Mg_{17}Al_{12}$ crystal. In both systems, the run driven by the self-consistent and smooth $\hat{\mathbf{F}}_c$ (blue) is the most stable, while the non-conservative $\hat{\mathbf{F}}_{nc}$ run (red) and the conservative but discontinuous run (green) are unstable.

**Temperature Fluctuations in NVT Simulations.** The non-conservative error of $\hat{\mathbf{F}}_{nc}$ also creates artifacts in NVT simulations. an NVT simulation of the liquid water box driven by $\hat{\mathbf{F}}_{nc}$ exhibits significantly larger temperature fluctuations (19.9 K stddev) compared to an equivalent simulation driven by the inaccurate self-consistent $\hat{\mathbf{F}}_c$ (10.6 K stddev) of the same `orb-v3-direct-inf-mpa`, even when using the same thermostat (Nose-Hoover with $\tau = 200$ fs). This demonstrates that the thermostat must work harder to counteract the unphysical energy being introduced by the non-conservative forces, which, as noted by Bigi et al. (2025), can disrupt the system's true structural and dynamical properties.

### A.2.2 DETAILED CORRELATION ANALYSIS: $U_\Delta$ AS A PROXY FOR $\varepsilon_c$ AND $\varepsilon_{nc}$

Table 2 provides the complete Spearman correlation results supporting the analysis in Section 4.2.1 and 4.2.2. It demonstrates the consistently strong correlation between $U_\Delta$ and $\varepsilon_c$ across the dynamic range, validating $U_\Delta$ as a proxy for the internal physical error. It also shows the strong correlation between $U_\Delta$ and $\varepsilon_{nc}$ during OOD exploration (adversarial attacks and NVE) and the use of $U_\Delta$ as a UQ metric for $\varepsilon_{nc}$ (section 4.2.2).

### A.3 DETAILED CORRELATION ANALYSIS FOR ENERGY DRIFT PREDICTION

This section provides the complete data and a more detailed analysis of the relationship between different error metrics and the total energy drift ($\Delta E_{\text{drift}}$) observed in NVE simulations. While the main text presents the key finding—that the internal inconsistency, $U_\Delta$, is a more consistent indicator of instability than the external error magnitude, $\varepsilon_{nc}$—this appendix details the model- and system-dependent nuances that support this conclusion.

As shown in figures 6 and 7, for the strongly-regularized `orbv3` models, we observe the most complex behavior. In this low-$\varepsilon_{nc}$ regime, the magnitude of the external error is a noisy and inconsistent predictor of drift. In contrast, the internal metrics, $\varepsilon_c$ and $U_\Delta$, maintain a more consistent positive correlation, making them more reliable indicators of the pathological character of the error that leads to instability. For the less-regularized `orb-d3-xs-v2` model, the system enters a more fragile state where $\varepsilon_{nc}$ becomes large, and as a result, all metrics ($\varepsilon_{nc}$, $\varepsilon_c$, and $U_\Delta$) become strongly correlated

Table 2: Spearman's rank correlation coefficients ($r_s$) between the Force Delta ($U_\Delta$) and the two error metrics ($\varepsilon_{nc}$ and $\varepsilon_c$).

| orb-v3-direct-inf-mpa (adv attack) | | | |
| --- | --- | --- | --- |
| System | Group | $r_s(U_\Delta, \varepsilon_{nc})$ | $r_s(U_\Delta, \varepsilon_c)$ |
| $Mg_{17}Al_{12}$ | Solid | 1.00 | 1.00 |
| LGPS | Solid | 1.00 | 1.00 |
| ice | Solid | 0.91 | 0.99 |
| MoF5 | Solid | -0.31 | 0.98 |
| CaPd-NH$_2$ | Surface | 0.58 | 1.00 |
| paracetamol | Molecule | 0.97 | 1.00 |
| stachyose | Molecule | 0.93 | 1.00 |
| Ac-Ala3-NHMe | Molecule | 0.95 | 1.00 |
| DHA | Molecule | 0.72 | 0.99 |
| aspirin | Molecule | -0.02 | 0.98 |

| orb-v3-direct-inf-mpa (NVE) | | | |
| --- | --- | --- | --- |
| System | Group | $r_s(U_\Delta, \varepsilon_{nc})$ | $r_s(U_\Delta, \varepsilon_c)$ |
| $Mg_{17}Al_{12}$ | Solid | 0.85 | 0.99 |
| LGPS | Solid | 0.92 | 0.93 |
| ice | Solid | 0.71 | 0.98 |
| Water | Liquid (Periodic) | 0.88 | 0.94 |

| orb-v3-direct-20-mpa (adv attack) | | | |
| --- | --- | --- | --- |
| System | Group | $r_s(U_\Delta, \varepsilon_{nc})$ | $r_s(U_\Delta, \varepsilon_c)$ |
| LGPS | Solid | 0.78 | 0.99 |
| ice | Solid | 0.96 | 1.00 |

| orb-d3-xs-v2 (adv attack) | | | |
| --- | --- | --- | --- |
| System | Group | $r_s(U_\Delta, \varepsilon_{nc})$ | $r_s(U_\Delta, \varepsilon_c)$ |
| $Mg_{17}Al_{12}$ | Solid | 0.73 | 0.94 |
| LGPS | Solid | 0.70 | 0.93 |
| ice | Solid | 0.70 | 0.99 |
| MoF5 | Solid | -0.24 | 0.05 |

| eqV2-dens-31M-mp (adv attack) | | | |
| --- | --- | --- | --- |
| System | Group | $r_s(U_\Delta, \varepsilon_{nc})$ | $r_s(U_\Delta, \varepsilon_c)$ |
| ice | Solid | 0.84 | 0.99 |
| $Mg_{17}Al_{12}$ | Solid | 0.42 | 0.98 |
| LGPS | Solid | 0.18 | 0.99 |
| CaPd-NH$_2$ | Surface | 0.23 | 1.00 |
| aspirin | Molecule | 0.44 | 0.99 |
| paracetamol | Molecule | 0.08 | 0.98 |

with each other and with the energy drift. Finally, for the `EquiformerV2` model, which has a strongly regularized $\hat{\mathbf{F}}_{nc}$ pathway, we observe that all metrics are again well-correlated, even though $\varepsilon_{nc}$ remains low. This complex landscape of correlations underscores the main conclusion: while the predictive power of $\varepsilon_{nc}$'s magnitude is model- and regime-dependent, the internal inconsistency, $U_\Delta$, serves as a more consistent indicator of algorithmic instability across these different scenarios.

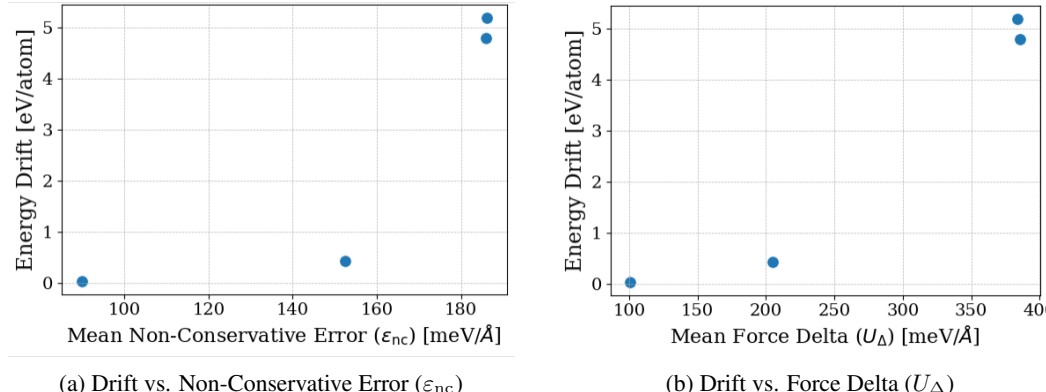

(a) Drift vs. Non-Conservative Error ($\varepsilon_{\rm nc}$)

(b) Drift vs. Force Delta ($U_\Delta$)

Figure 6: The Force Delta ($U_\Delta$) as a consistent indicator of algorithmic instability for `orb-d3-xs-v2` models. Each point represents a 10-ps NVE simulation for each system. (a) In the low-error regime, the magnitude of the non-conservative force error (b) In the same set of simulations, the internal inconsistency

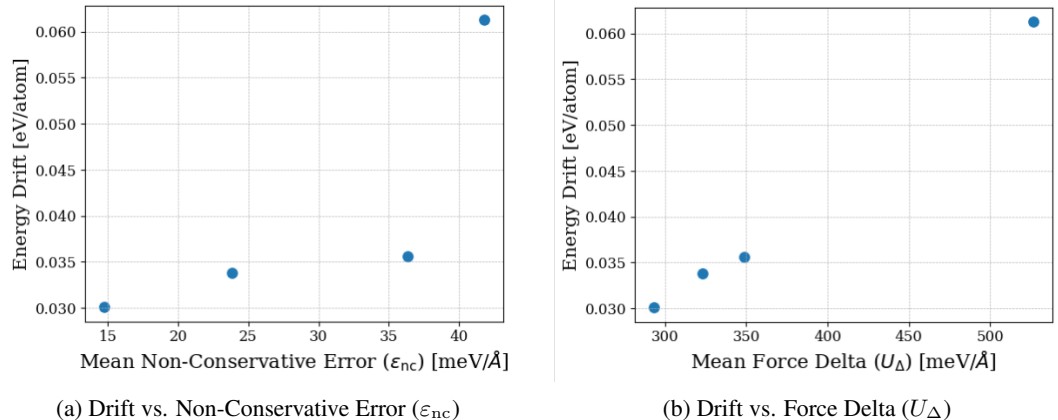

(a) Drift vs. Non-Conservative Error ($\varepsilon_{\rm nc}$)

(b) Drift vs. Force Delta ($U_\Delta$)

Figure 7: The Force Delta ($U_\Delta$) as a consistent indicator of algorithmic instability for `eqV2-dens-31M-mp` models. Each point represents a 10-ps NVE simulation for each system. (a) In the low-error regime, the magnitude of the non-conservative force error (b) In the same set of simulations, the internal inconsistency

### A.4 ADDITIONAL FINE-TUNING RESULTS

To demonstrate the generality of the fine-tuning results presented in Section 4.4, we performed an equivalent experiment on a different model and system: the `orb-d3-xs-v2` model on the LGPS crystal. As shown in Figure 8, we observe the same stepwise improvement in stability. The pre-trained model exhibits significant energy drift. Fine-tuning on 100 adversarial data points on the $\hat{\mathbf{F}}_{\rm nc}$ head alone reduces the drift, and fine-tuning both the $\hat{\mathbf{F}}_{\rm c}$ and $\hat{\mathbf{F}}_{\rm nc}$ heads further improves the stability of the $\hat{\mathbf{F}}_{\rm nc}$-driven simulation.

Figure 8 shows the full energy evolution trajectories for the NVE simulations of both the ice and LGPS systems. The plots clearly illustrate the reduction in both short-term drift and long-term instability at each stage of the fine-tuning process.

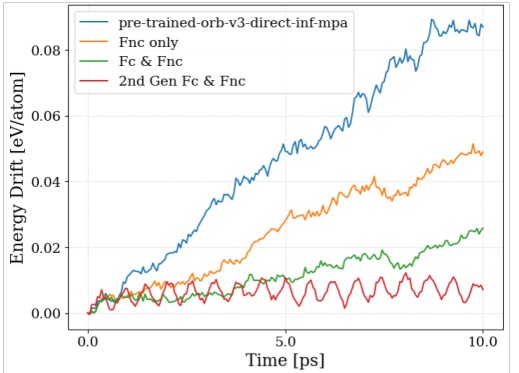 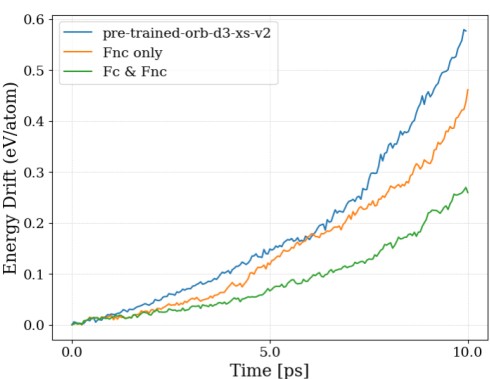

Figure 8: Energy drift during NVE simulations for the pre-trained and fine-tuned models. (a) The `orb-v3-direct-inf-mpa` model on the ice system. (b) The `orb-d3-xs-v2` model on the LGPS system. Each stage of fine-tuning leads to a more stable trajectory with reduced energy drift over time for both systems.

