# OpenReview forum: "When Forces Disagree: A Data-Free Fast Diagnostic from Internal Consistency in Direct-Force Neural Network Potentials"
_ICLR.cc/2026/Conference — ICLR 2026 Conference Desk Rejected Submission_

### Official Review · Reviewer_VVBh · 2025-10-26

**Soundness:** 2
**Presentation:** 3
**Contribution:** 2
**Rating:** 2
**Confidence:** 4

**Summary:**

This paper proposes the Force Delta as a measure of the consistency between the conservative and non-conservative forces predicted by a machine-learned interatomic potential equipped with an additional direct-force head that predicts non-conservative forces. Mathematically, the Force Delta is the root mean square deviation between the conservative and non-conservative forces predicted by the model on a given atomic structure. The authors use this metric:
- to establish the claim that internal consistency of the two force predictions is more important than the accuracy of non-conservative forces as a predictor of stable simulations
- as a proxy for the error of both force types, effectively using it as an uncertainty quantification metric for forces
- to generate new structures for fine-tuning the model and improve its accuracy and behavior in simulations

**Strengths:**

**Originality**

The content is fully original. It builds an interesting and novel idea based on recent literature in the field. The "adversarial" generation of structures for fine-tuning using the Force Delta is especially interesting and worth pursuing in my opinion.

**Quality**

The proposed investigation of the Force Delta is thorough and of good standards, despite the fact that the authors might have missed some points in their analysis.

**Clarity**

The paper is readable. In particular, its sections are structured in a reader-friendly way.

**Significance**

Although not a groundbreaking advance, the paper investigates a novel idea with the potential for broad interest in the field. Despite the fact that I disagree with the authors' point of view regarding whose responsibility it is to provide UQ diagnostics for simulations (see below), I believe that practitioners could reasonably and easily use the Force Delta metric as a useful tool in their work.

**Weaknesses:**

Here, I believe it is useful to make a distinction between different degrees of weaknesses, and I will therefore divide them into three categories.

**Major -- in my opinion preventing publication in the current state**

- The authors claim that "model ensembles are data-free but computationally expensive" but that "faster single-model methods often require access to the original training data". Creating an ensemble of models clearly requires the training data as well. Unfortunately, I believe that the authors are treating the two on two different grounds. Either (1) the authors expect the creators of the model to build the ensemble, while they expect the creation of a "single-model" uncertainty method to be a responsibility of the user, or (2) the authors believe that an ensemble of fine-tuned models can be created by practitioners, while the same thing is not possible for "single-model" methods. In reality, as a counter-argument to (1), since model creators have access to the training data, they can (and it should be their responsibility to) provide "single-model" UQ capabilities. This is not an unrealistic expectation (it has been done, for example, for the MACE-MP and PET-MAD universal potentials). As a counter-argument to (2), single-model UQ can be performed on fine-tuned models using only the fine-tuning data. This is equivalent to treating the pre-trained model as a Bayesian prior, and it is exactly the same that one does in practice when creating an ensemble of fine-tuned models. I do believe that the Force Delta might, in practice, still present advantages over single-model force UQ, but these would be due to practicality and neglect of UQ by the model creators rather than methodological advantages.

- The authors claim that "the relative importance of conservativity compared to accuracy has not been directly demonstrated" (line 54). This claim is incorrect. One could argue about what "directly" means here, but several papers cited by the authors (as well as public benchmarks) show a wide range of examples of conservative models performing better than more accurate non-conservative models on several downstream tasks. I will be happy to provide several examples in the rebuttal phase, if needed. The authors only provide an additional (and trivial) example, which is, in passing, also provided in the same form (inaccurate conservative model conserving energy vs inaccurate non-conservative model drifting) by their references and public benchmarks.

- The authors claim that "the current state of the art is dominated by E(3)-equivariant GNNs" (line 97), while then using ORB for most of their experiments. The authors should then be aware of the fact that ORB is a non-equivariant state-of-the-art model and that it is more accurate and faster (given the current evidence) than most of the equivariant architectures that the authors cite as a support to their claim. Besides ORB, several other state-of-the-art non-equivariant architectures are available, and I would be happy to provide a few examples during the rebuttal phase, if needed.

- The authors' analysis is based on the models being trained with the loss in Eq. (1), which does not include backpropagated forces. This is however not considered to be a good way to train models anymore. Models trained with this loss perform poorly on public benchmarks compared to their conservative equivalents and are rarely used by practitioners. Model creators now prefer conservative training (and this includes the creators of Equiformer and ORB, which are used in the manuscript). [For completeness, the recently established state-of-the-art for large-scale model training seems to be non-conservative pre-training followed by conservative fine-tuning (Fu et al 2025, Bigi et al 2025).] The same analysis, where the Force Delta is shown to be dominated by the error of the conservative forces, would not hold for models whose conservative force predictions are also trained. In fact, these models tend to extrapolate better than non-conservative models due to their more correct physical prior.

**Minor**

- The authors cite MACE as an example of a model using non-conservative predictions (line 106). However, to the best of my knowledge, no MACE models with non-conservative heads exist and the MACE creators are opposed to non-conservative predictions. I invite the authors to correct me if I am mistaken, or to limit their citations to Gemnet, Equiformer, ORB and potentially other more recent models (I can provide more examples if needed).
- "Symplectic integratiors used in MD assume forces are the exact gradient of a potential to conserve the Hamiltonian" (line 128) and "satisfies the symplectic requirements of the integrator by being an exact gradient of its own predicted energy" (line 221). While these statements are correct, it is important to note that poor behavior in molecular dynamics is not the integration algorithm's fault. The Hamiltonian of non-conservative force models is simply not defined, and no integration algorithm would be able to make the dynamics Hamiltonian and therefore well-behaved.
- "The limitations of existing UQ models" section: there exists a large amount of literature for UQ on atomistic systems (including fast UQ models that are have been incorporated in universal interatomic potentials) that the authors are ignoring. However, I would understand if, due to the page limit, the authors did not manage to include the relevant citations in the main text. In that case, a more thorough discussion should be added in the appendices and made available in the main text in versions of the manuscript where the page limit is lifted.
- "However, they represent the most direct ensemble-based UQ approach available to a user working with publicly available pre-trained models." (line 314) Here, the authors use an ill-defined ensemble to quantify the uncertainty of the models. Once again, I think it would be the model creators' responsibility to provide good UQ methods, instead of the users having to put together a pathological ensemble of different models trained with different architectures and hyperparameters, and on different levels of DFT. The authors could instead use the errors on the forces to find a correlation to the Force Delta, with a slightly different effect but likely allowing to reach the same conclusions.

**Typos**

- "unstability" -> instability (line 52)
- "mostly recover" -> mostly recovering (line 44)
- "real-time" -> real time (line 46)
- "are" -> is (line 53)
- comma missing at the end of line 184
- "produces artifact" -> produces this artifact (?) (line 220)

**Questions:**

See "weaknesses" for some points of discussion. Excluding those, I only have one further question, which other reviewers might find appealing as well and reply to, if they see fit.

If the Force Delta were to be evaluated on models whose energy output is also trained using back-propagated forces, do the authors think the same UQ properties would hold? In that case, although the current analysis might not be valid anymore, the conservative and non-conservative predictions coming from the same model could be seen as an ensemble with two members, which would therefore still be able to quantify the predictive uncertainty to an extent.

---

### Official Review · Reviewer_upzb · 2025-10-29

**Soundness:** 3
**Presentation:** 3
**Contribution:** 3
**Rating:** 6
**Confidence:** 4

**Summary:**

The paper discusses how consistency between direct force predictions and predictions computed as the derivative of a potential can serve as a proxy for predictive uncertainty. A simple metric is defined and several experiments demonstrate its practical usefulness. A fine-tuning experiment using an adversarial sampling strategy shows how direct and conservative force predictions are linked through their shared representation.

**Strengths:**

The paper is well written and easy to follow.

The paper explores the simple and useful idea of using the difference between conservative and non-conservative force predictions as an uncertainty proxy. Several experiments demonstrate that this is practically useful.

**Weaknesses:**

The key idea is simple but not particularly groundbreaking. For example, several earlier studies have used stability of MD simulations to assess the quality of direct force predictions, and the step to the proposed metric is not so large.

The cited related work on single-model UQ methods is limited.

There is no direct comparison with deep ensembles or single-model UQ methods. Thus, while the paper clearly demonstrates the usefulness of the proposed metric, it is not clear how other approaches compare. The only direct comparison is with an ensemble of pretrained, publicly available models, and it is not clear to me that this necessarily leads to good UQ performance.

**Questions:**

Is there a missing "interatomic" in the first sentence in the abstract?

By "algorithmic stability" in the abstract, I assume you mean e.g. in MD simulation? Maybe this could be more clearly stated here.

"(...) faster single-model methods often require access to the original training data." Could you mention some of these and why they require access to training data at inference time? Also, there are many single-model methods that do not require access to training data, so is this not a bit misleading?

It is a bit unclear what it means that force delta is "data-free" - surely it is computed for a particular configuration.

"The first head predicts the direct, non-conservative force (...)" This sounds like the target force is not conservative.

What exactly is the notation for the atom positions? Is R the collection of all positions in the system? What is lowercase r in eq. 3 then?

Is it at all surprising that self-consistency is more important for stability than small error, given that the simulation will accumulate such error? I assume a conservative force field will only lead to instability if the potential energy surface from which it is computed is degenerate in some way.

"Bigi et al. (2025) have demonstrated that this artifact in NVT is difficult or impossible to contain using a thermostat without disrupting dynamical properties." I understand the point, but is "difficult or impossible" not a bit of an over-statement? Surely, in many cases when non-conservative force predictions are accurate enough, suitable thermostatting is enough to stabilize the simulation?

"In contrast, the model’s energy, Ê, is only weakly supervised by scalar values (...)". Typically, a the energy function is trained on both target energies and forces from DFT, i.e. the loss function includes an energy term and a force term fitted to the gradient of the predicted energies. What is meant by "weakly supervised"? Do you mean a setting where the energy is trained only on energy targets?

What are the details on the ensembles of pretrained models used in table 1? This information seems to be missing from the main text.

Why did you not compare with a deep ensemble? This seems to be the best know method for UQ, so why not do a direct comparison?

"While the accumulated error, εnc (...)" Is this the accumulated error, not just the non-conservative force error? Did you mean to say "the accumulation of the error εnc" or something like that?

How exactly is the fine-tuning experiment carried out? When fine-tuning F_c I assume you fine-tune the entire architecture (and not just the head) such that the learned representation will change? I assume you fine-tune with a loss that combines energies and forces? When fine-tuning F_nc I assume you fine-tune only the head?

---

### Official Review · Reviewer_FpyM · 2025-10-31

**Soundness:** 3
**Presentation:** 3
**Contribution:** 3
**Rating:** 6
**Confidence:** 2

**Summary:**

This paper introduces the "Force Delta"  a data-free uncertainty quantification metric for direct-force neural network potentials. The core idea is to leverage the internal disagreement between a model's direct force prediction and its energy-derived conservative force as a diagnostic for model reliability and simulation stability.

**Strengths:**

The proposed method is practical, computationally cheap, and has the potential for immediate impact.

- the idea of using the internal inconsistency of a dual-output model as a UQ metric is elegant and powerful
- comprehensive validation
- the paper is well-structured and clearly written

**Weaknesses:**

- The term "data-free" is used prominently, but it's slightly nuanced. The metric itself is data-free at inference time, which is its major advantage. However, the validation of the metric and the fine-tuning workflow still require DFT data. This should be clarified.

- more discussion about the limitations

**Questions:**

- The fine-tuning results are impressive. Did you observe any trade-off between improved stability and the static accuracy (on a standard test set) of the model after fine-tuning on U-maximized adversarial data?

- The inter-head influence is attributed to the shared GNN embedding. Did you consider or perform any ablation studies (e.g., using a model with less shared parameters) to further isolate the effect of the shared representation on the correlation measured by U?

- The adversarial attack minimizes energy while maximizing U. How sensitive are the results to the balance between these two objectives (the α and β parameters in Eq. 3)? Is there a risk of generating physically unrealistic configurations?

---

> ### Author Response · Authors · 2025-12-04
>
> We appreciate the reviewer's feedback.
> - We agree with the reviewer and have clarified that Ud is data-free at inference time.
> - We have added more content into the limitation section (e.g., Ud's reduced efficiency since conservative forces calculation also incurs extra computational cost) as suggested by the reviewer
> - We did not observe any correlation or trade-off between improved stability and the static accuracy on held-out test set in the finetuning experiments since the accuracy is mostly high after finetuning. This further strengthens the point that pure static accuracy is not enough for stable simulations.
> - We agree with the reviewer and further performed and compared the results on small orb and large orb models, which have small and large shared parameters between heads, respectively. We find that the correlation is still large for both cases, indicating that the cause the existence of the shared representation itself rather than the size of the representation.
> - Yes, the reviewer is totally right. If the energy weight is too small, the adversarial attack process could generate unphysical configurations with high Ud and high energy.

---

### Official Review · Reviewer_B1Hy · 2025-10-31

**Soundness:** 3
**Presentation:** 2
**Contribution:** 3
**Rating:** 8
**Confidence:** 4

**Summary:**

This work deals with the question of assessing prediction uncertainty in machine-learning interatomic potentials (MLIPs), and in particular those trained to predict forces directly, rather than as gradients of a scalar potential energy $E$. Such forces do not respect energy conservation and are therefore called non-conservative, and can yield unphysical results when used for simulations. Predicting their uncertainty is therefore a pressing requirement to control error in practical simulations.

To this end, the authors introduce the force disagreement $U_{\Delta}$, defined as the difference between the non-conservative forces predicted by the model and the conservative forces obtained by differentiating the energy prediction of the same model. They find that this error is a good predictor for simulation instability, and is superior to measuring error against ground truth evaluated on a test set. They provide some theoretical justification for the effectiveness of the metric: The derivatives of the scalar energy are poorly supervised during non-conservative training, leading to strong discrepancies out-of-distribution compared to the well-supervised trained forces. Finally, they show that the energy and force prediction heads in non-conservative MLIPs are not independent and that improvements of the potential energy prediction (and its derivatives) also improves the physicality of the non-conservative forces.

Note: An LLM (ChatGPT 5) was used in proofreading and formatting this review. It also suggested some points in the review, few of which were included in the suggestions (expanded OOD tests/units & thresholds/varying energy & force weights).

**Strengths:**

This paper is a strong contribution to the developing literature on non-conservative MLIPs. The proposed uncertainty measure is simple, yet novel, and appears to be practically useful. The experiments support the claims and are reasonable. The paper is written clearly.

**Weaknesses:**

The paper could be improved to make its contributions more clear, position them better in the literature, and make the experimental evidence stronger. The applicability of $U_{\Delta}$ is somewhat unclear and the claims of much improved efficiency with respect to other UQ methods should be expanded upon. Some experiments lack a stronger UQ baseline or could be improved with additional metrics.

I have expanded on these weaknesses in the question and suggestion section, and will be happy to raise my score if they are addressed.

**Questions:**

- As $U_{\Delta}$ relies implicitly on the lack of supervision for the scalar energy head, does this imply that the metric is applicable exclusively to models trained non-conservatively? If yes, this should be stated clearly.
- Consequently, does conservative fine-tuning degrade the predictiveness of the metric? This question should be easily answered through the experiment in section 4.4. Please report the correlation of $U_{\Delta}$ with instability pre- and post fine-tuning.
- Regarding the use of $U_{\Delta}$ in monitoring running MD, does the metric precede emerging instability? I.e., can it catch failures before they happen?
- As computing $U_{\Delta}$ requires access to the conservative forces, it negates the speed advantage of non-conservative forces. This should be stated clearly, as it is a (non-critical) shortcoming of the approach.
- In section 4.3, I would be curious to learn if $U_{\Delta}$ correlates well with the asymmetry of the Jacobian that was proposed as non-conservativity metric by Bigi et al.
- The literature review should be expanded, as there has been much work on UQ methods that are more efficient than full-model ensembles. For instance: Kellner et al. (DOI:10.1088/2632-2153/ad594a) and references therein, Swinburne & Perez (DOI:10.1088/2632-2153/ad9fce), Perez et al. (arXiv:2502.07104), Bigi et al. (DOI:10.1088/2632-2153/ad805f).
- In light of this expanded literature section, I would suggest less strong claims about the efficiency relative to other UQ methods, in particular if those are not benchmarked explicitly. As conservative forces also incur extra computational cost on the order of 2-3x the forward pass, I don't see how $U_{\Delta}$ is more efficient than a very small ensemble. An explicit benchmark would make this claim stronger.
- The results presented in section 4.2.2 could be made much stronger by comparing with a stronger UQ baseline, for instance a (small) full or shallow ensemble of models, or the last-layer approximation in Bigi et al. cited above.
- As both $U$ and $\epsilon$ are not uncommon as symbols for the potential energy in the literature of atomistic simulations, please consider changing the notation.
- I am personally unfamiliar with the term "algorithmic instability". Could you please define it, and clarify why you chose it instead of "simulation instability" or a similar term?
- Have you considered OOD tests that are not adversarial, but rather more aligned with typical MLIP usage? I.e., the formation of defects in solids, application of strains, or bond-breaking. Does $U_{\Delta}$ remain predictive in this regime?
- Please state the units of $U_{\Delta}$ and suggested thresholds for practical use for UQ in the text.
- If you have the bandwidth for additional experiments, the claim that regularization is the driving factor in the effectiveness of $U_{\Delta}$ could be made stronger by systematically varying the weight of energy and forces during training.

While reading the manuscript, I have noted the following small errors that should be fixed for the camera-ready version (they have not impacted my overall assessment):

- Line 174: "trained" is misspelled
- Line 220: "artifacts" is misspelled
- Line 729: "An" should be capitalized
- Figures 6 and 7: The captions appear to be truncated

---

> ### Author Response · Authors · 2025-12-04
>
> We thank the reviewer for reading the paper and providing multiple constructive feedback.
> - Yes, our results are exclusively applicable to models pretrained non-conservatively. We intended the metric to be used for fast improvement and monitoring of open-source pretrained models like Orb which are impactful since they are easy to use out-of-the-box for many end users, to encourage more adoption because of their speed advantage (up to 4-5 times faster than conservative forces) which is impactful for MLIP practitioners. Conceptually, however, we believe Ud should also be applicable to models with both forces pretrained since if both forces disagree, at least one of them is inaccurate, and through the inter-head influence we have demonstrated in the paper, the other force is also likely to be inaccurate. Especially for the MTS solution suggested by Bigi et al., where both forces are used in the simulation, if at least one of them is inaccurate the simulation is no longer reliable. We have added this information in the limitation section. Thank you very much.
> - Following the reviewer's suggestion, we compute the average Ud along the simulation trajectory for each finetuned model and calculate the correlation with total energy drift at the end of the simulation. Ud still correlates well with the instability, indicating that it is still predictive even for conservative finetuned models, which we believe strengthens our paper.
> - We find that Ud's magnitude moves simultaneously with the instability during NVE, which can be easily monitored from the total energy predicted by an MLIP. However, as shown by Bigi et al. that this instability can cause more issues to dynamical and structural properties not predicted on-the-fly by the MLIP being used in non-NVE setting, we believe Ud is still useful for monitoring non-conservativity artifact in those cases.
> - Following the reviewer's suggestion, we have added this point, which we agree with, to the limitation section. Thank you.
> - Conceptually, we believe Ud should correlate well with the asymmetry of the Jacobian since the conservative force in Ud will have zero asymmetry. Therefore, the distance of the direct force from the conservative force as measured by Ud should reflect the degree of Jacobian's asymmetry. We apologize for not including the evidence due to the time limit since Jacobian is very expensive computationally, which supports our Ud as a fast alternative for monitoring non-conservativity during the simulation.
> - We agree with the reviewer; subsequently, we have added more literature on UQ methods as suggested.
> - We agree and have modified our claims to make it less strong as suggested by the reviewer. As we wanted to make this paper impactful by focusing mainly on open-source large-scale models pretrained on large-scale data, which are more impactful since they are accessible to large audience, we believe a very small ensemble baseline would be a bit irrelevant here since it is rarely used in practice by MLIP practitioners. Instead, we choose an ensemble of pretained models which still share the same training dataset (mptraj). We believe this pretrained ensemble is still valid as it still reflects the key idea of deep ensemble uncertainty which is the disagreement between multiple models trained on the same dataset. Moreover, this pretrained ensemble is much more relevant to end users since they can calculate right out of the box from available open-source models rather than having to train deep ensembles on large-scale datasets. Training deep ensembles on large data from scratch for end users is highly impractical, time-consuming, and in turn defies the intended fast access to open-source pretrained models .
> - We agree with the reviewer and we have subsequently added a GMM baseline as a single-model uncertainty (as proposed by 10.1063/5.0136574) using a subset of mptraj data that includes elements of the four solids intended as in-distribution in our test set to fit the GMM, we found that the GMM uncertainty performs slightly worse than both Ud and ensemble, strengthening our Ud as an uncertainty metric that combines both advantages of ensembles and GMM by being data-free at a single-model speed.
> - We agree and we will change the notation in the final version of the manuscript.
> - We apologize for the uncommon and undefined phrase. We have changed the term to "simulation stability" as suggested by the reviewer since it reflects what we wanted to communicate.
> - Yes, the current non-adversarial test set also includes surface and molecules which are OOD since they are not in the training dataset for the pretrained models used in this paper (mptraj)
> - We agree to the reviewer's point and have added it as future directions in the paper as the rebuttal period is too short for those experiments unfortunately. We sincerely hope the reviewer understand.
> - We also thank the reviewer for catching small errors.

---

### Note · Program_Chairs · 2026-01-17
**Submission Desk Rejected by Program Chairs**

The following references in this submission do not refer to real documents and/or have major errors in bibliographic information:

 Cas van der Hirschfeld, Giulio Imbalzano, and Michele Ceriotti. Uncertainty quantification in atomistic machine learning. The Journal of Chemical Physics, 153(10), 2020.
Yiqi Wang, Xinyue Wang, Raymond A Adomaitis, and Dongxia Liu. Rethinking the implementation of an uncertainty-aware deep learning framework for materials property prediction. Digital Chemical Engineering, 7:100097, 2023.